# Identity construction in the very old: A qualitative narrative study

Helen Anderson[1]*, Rachel Stocker[2], Sian Russell[2], Lucy Robinson[2,3], Barbara Hanratty[2], Louise Robinson[2], Joy Adamson[1]

1 York Trials Unit, Department of Health Sciences, University of York, Heslington, York, United Kingdom, 2 Population Health Sciences Institute, Faculty of Medical Sciences, Newcastle University, Newcastle Biomedical Research Building, Campus for Ageing and Vitality, Newcastle upon Tyne, United Kingdom, 3 Northumbria Healthcare NHS Foundation Trust, Cobalt Business Park, Newcastle Upon Tyne, Tyne and Wear, United Kingdom

* helen.anderson@york.ac.uk

**Data Availability Statement:** This paper reports on a qualitative data set for a very specific group of participants (very old people aged between 97-99 years in a specific geographical setting) who could be identifiable, and participants did not expressly

## Abstract

People are living longer internationally, with a growing number experiencing very old age (≥95 years). Physical, psychological and social changes can challenge one's sense of self and disrupt existing identities. However, experiences of the very old in society are seldom researched and how they construct identity and negotiate a sense of self is little understood. Our study focuses on participants aged >95 years to understand how identity is conceptualised to negotiate a continued place in society. Qualitative interviews with 23 people were thematically analysed, underpinned by Positioning Theory. Five themes were generated: A contented life; reframing independence; familial positioning; appearance and physical well-being; reframing ill health. Participants saw themselves as largely content and, despite their world becoming smaller, found pleasure in small routines. Perceptions of self were reframed to maintain autonomy within narrow parameters. Past relationships and experiences/events were drawn on to make sense of ongoing ways of living. There were tensions around feelings of loss of autonomy and independence, with some valuing these over issues such as safety. This sometimes conflicted with views of others and small acts of resistance and subversion were acted out to maintain some sense of control. However, participants minimised progressive ill health. Findings provide insight into how the very old may utilise identity to negotiate, acquiesce, resist and challenge the world around them.

## Introduction

More people are living longer internationally [1, 2] and those aged ≥85 years are the fastest growing population group in the United Kingdom (UK) [3]. It is anticipated that age at death will continue to increase in the UK [3] with a growing number of people experiencing very old age (≥95 years) [4], although the impact of the Covid-19 pandemic is unlikely to be clear for some time [5]. However, experiences of the very old in society are seldom researched [4].

Ageing has been traditionally conceptualised by withdrawal from society and characterised across social and medical discourses as a period of decline. This presents older people as a

consent to their data being shared publicly. Therefore, there are ethical restrictions on the data set. While the data has been pseudonymised, the detailed nature (and select cohort) is such that it cannot be fully de-identified and consequently, is only available on request as per ethical approval from the Newcastle and North Tyneside 1 NRES committee (ref: 06/Q0905/2) (approved 2006), and subsequent substantial amendment approved by the North East - Newcastle & North Tyneside 1 Research Ethics Committee (Substantial Amendment no.23, 16th July 2019). Anonymised excerpts of the transcripts from the qualitative interviews are reported within the paper. Pseudonymised transcripts are available via the University of Newcastle Repository. Requests can be made to the Newcastle 85+ Data Repository via https://research.ncl.ac.uk/85plus/datarequests/. Formal requests for access will be considered via a data sharing agreement that indicates the criteria for data access and conditions for research use and will incorporate privacy and confidentiality standards to ensure data security.

**Funding:** This project is funded by the National Institute for Health Research (NIHR) under its Research for Patient Benefit (RfPB) Programme (Grant Reference Number PB-PG-1217-20025). The views expressed are those of the author(s) and not necessarily those of the NIHR or the Department of Health and Social Care. Authors awarded funding were LR, BH and JA. All researchers were independent from funders. The funders had no role in study design, data collection and analysis, decision to publish, or preparation of the manuscript.

**Competing interests:** The authors have declared that no competing interests exist.

burden and perpetuates ageism [6]. Satisfaction with life has been found to decline rapidly in those aged over 74 years, reaching its lowest level in those aged 90–94, with life satisfaction associated with perceived health [7]. Similarly, in individuals aged 78–98 years, the number of symptoms experienced has been found to negatively impact on life satisfaction [8]. External societal constructions of ageing often portray ageing negatively and position an increasing age-ing population as a risk to health, social and government services [6]. Consequently, ageing is stigmatised [9] and the focus on biomedical and economic deficit aspects of ageing neglects the historical, political and socio-cultural construction of ageing and limits critical engagement with ageing issues [10]. Consequently, alternative and additional ways of exploring ageing are necessary. One such way is the exploration of identity work in those experiencing very old age.

Age identity is narrated and played out throughout the life span [11] and ageing can pose a threat to identity [12]. Changes in physical, psychological and social roles can challenge one's sense of self and cause disruption to existing identities in later adulthood [13]. Identity con-struction in older age is increasingly considered to be a lifelong ongoing process which draws on conceptualisations of past, present and future and is influenced by socio-cultural factors [6]. Indeed, ageing, and life course transitions, are argued to be embodied within the individ-ual and negotiated in everyday life through the interaction between aspects of lived experience such as health and identity [10]. By exploring identity narratives within the context of lived experiences and individual biographies, new insight into the lives of the very old can be devel-oped. However, despite the increased number of people achieving this age [1]. experiences of the very old in society are rarely researched [4, 14] and how they construct their identity and negotiate their sense of self in their world, especially as they age [14], is little understood. Our study focuses on participants aged over 95 years. Therefore, this is one of very few studies focusing of this age group and affords a unique opportunity to gain insight into ongoing iden-tity formation. The aim of this study is to give voice to, and gain understanding of, how some of the oldest people in society (aged over 95 years) conceptualise their identity and negotiate their continued place in the world.

## Positioning theory

The study draws on Positioning Theory, which was developed by Davies and Harré [15]. Iden-tity is understood as being narratively constructed by the individual and by the relationships and interactions they develop within wider social, historical, political and cultural contexts. Reflecting the work of Goffman [16], individuals seek to manage the impressions they give to others through the performance of social interactions and, in this way, construct and maintain a sense of self. Positioning Theory is derived from social psychology and is underpinned by a social constructionist approach. Individuals and groups are positioned by themselves, by oth-ers and by wider society and the positions assigned influence how a person understands them-selves, how they are perceived by others and how they behave [17]. Identity is continuously created through narrative plotlines which are developed, organised and omitted in order to give meaning to events [18]. In this way we use narratives to construct our identity in the world [19]. A key principle of Positioning Theory is that identity is created through the inter-action between the stories people tell themselves and others within local contexts [20] and broader social, cultural, historical and political discourses [17], which have been termed 'mas-ter-narratives' [p89]. Examples of master-narratives would be societal perceptions of what it is to be a teacher/student or a doctor/patient. Individuals draw on master-narratives to position themselves and others within storylines in order to claim rights and duties for themselves and to ascribe and challenge rights and duties to others [17, 18]. For example, a teacher may claim certain rights over their pupils, while accepting certain responsibilities. Dominant master-

narratives of ageing are often associated with negative stereotypes such as dependence on others and physical and cognitive limitations. Therefore, we can gain a greater understanding of identity work in the very old by considering how underlying master-narratives may influence the positions older people take/are attributed and the contribution this makes to formulating identity [9]. However, while master-narratives are often tacit and undisputed [21], they can also be contested and rejected [22]. In this way individuals are not passive, but have agency and are active in their identity construction [22]. Furthermore, identity construction is seen as a non-linear iterative process [17] and can be constructed not only through autobiographical-type narratives, but also in what are termed small stories [23]–everyday, fleeting, mundane conversations and stories which are sometimes not fully formed and may lack coherence as a narrative. The naturalistic nature of such narratives is central [22].

A model of Positioning Theory has been developed to interpret narratives and form analytical steps [17, 22–24]. Bamberg's three level positioning analysis explores identity construction by using narratives told by individuals [22]. At the first level of analysis, we identify how the individual positions themselves as a character in relation to other characters in the narrative. It then considers how the narrator positions him/herself in relation to the audience to whom they are telling their story. This is because when individuals present themselves to their audience (e.g. the researcher or the wider audience of the study) this represents choices about how they wish to be seen. It also reflects culturally embedded identity narratives. At the third level of analysis, the focus is on how the narrator positions themselves in terms of past events and pre-existing master-narratives in order to achieve a sense of self and identity i.e. how the narrative fits with wider societal and cultural discourses [17, 21, 24]. For Bamberg [22] this third analytical level highlights how, and in what ways, individuals construct 'who they are' [p337]. In this way identity can be explored as interactive, cognitive, social and structural processes. Of note is that there is a dynamic relationship between the levels and, consequently, the levels are cross cutting and intertwined. For example, when constructing an identity of 'mother' an individual will consider how they position themselves as a mother in relation to other characters in the narrative they tell (level 1), how they want to present themselves as a mother to the audience (level 2) and this will be underpinned by their understanding of socio-cultural master-narratives about what it is to be a mother (level 3). It is through the interaction between these layers of narrative that we make sense of our place in the world and formulate who we are. As a consequence, Positioning Theory provides both a theoretical framework (lens) through which to explore how identity is narratively constructed and presented through small stories and everyday conversation, as well as an analytical tool to aid understanding of how the very old perceive identity and carry out identity work. In this paper we use Positioning Theory as a lens through which to understand how individuals used the stories they told to construct and present their identities.

## Methods

### Study design

A qualitative interview study was designed to gain, in-depth, participants' perspectives and experiences of living life as the very old in society and the resulting challenges and advantages, set within the context of their individual biographies. To this end we specifically aimed to explore, through a Positioning Theory lens, how self-concept and identity is experienced.

### Participants

This paper reports on findings from data generated through qualitative interviews with people aged between 97–99 years old (n = 23) (Table 1). Participants were purposively sampled from

**Table 1. Participant information and interview schedule.**

| Participant Pseudonym | Interviewed in 2018/19 | Interviewed in 2020 |
|---|---|---|
| Russell | ✓ | |
| Cath | ✓ | |
| Judith | | ✓ |
| Tony | ✓ | |
| Bob | ✓ | |
| Mary | ✓ | |
| Eileen | ✓ | |
| Adele | ✓ | |
| Pauline | ✓ | |
| Gwen | | ✓ |
| Maureen | ✓ | ✓ |
| Jack | ✓ | ✓ |
| Alan | ✓ | |
| Pamela | ✓ | |
| Carol | ✓ | |
| Anne | ✓ | |
| Penny | ✓ | |
| Angela | ✓ | |
| Margaret | ✓ | |
| Joe | ✓ | ✓ |
| Malcolm | ✓ | |
| Jean | ✓ | |
| Susan | | ✓ |

the 80 remaining participants in a 10 year follow up study to The Newcastle 85+ Study [14, 25] which is a longitudinal observational cohort study of people born in 1921 and who were registered with Newcastle upon Tyne or North Tyneside Primary Care Trusts in the UK in 2006 [http://dx.doi.org/10.17504/protocols.io.10.1186/1471-2318-7-14]. Sampling attempted to ensure variation within the confines of the study population: gender (female n = 16, male n = 7); stated place of residence (where specified) at age 95 (private residence n = 15; sheltered housing n = 3; residential care home n = 1; other n = 1); living independently (n = 14); living with partner (n = 3); living with son/daughter (n = 3); morbidity and frailty. Participants had a range of health and social care needs and different levels of support from family and friends. In order to contextualise the potential life experiences of people born in 1921 and living in the UK, a general timeline of key events has been provided (Box 1). Study participants would have reached young adulthood at the time of World War 2.

Ethical approval for the original 85+ study was obtained from the Newcastle and North Tyneside 1 NRES committee (ref: 06/Q0905/2) in 2006, and a substantial amendment to carry out the work described in this paper was approved by the North East—Newcastle & North Tyneside 1 Research Ethics Committee (Substantial Amendment no.23, 16th July 2019). Participant information leaflets were provided and written informed consent was obtained from all participants. A member of the main study's public involvement group was involved in the study design, co-developed participant information leaflets and contributed to the interview topic guide.

## Box 1. Contextual timeline for study participants.

| Date | Contextual Timeline in relation to Study Participants | Age of Study participants | Source |
|---|---|---|---|
| 1914–1918 | World War 1 resulted in 880,000 British deaths, 6% of the adult male population and 12.5% of those serving. The toll on the adult male population meant that the 1921 Census recorded 109 women for every hundred men. | | https://www.parliament.uk/business/publications/research/olympic-britain/crime-and-defence/the-fallen/#:~:text=In%20WWII%20there%20were%20384%2C000,half%20of%20them%20in%20London. |
| 1918–1920 | Influenza pandemic "Spanish flu". Killed 250,000 people in Britain | | https://www.redcross.org.uk/stories/health-and-social-care/health/coronavirus-how-the-red-cross-helped-in-the-spanish-flu-pandemic |
| 1916–22 | David Lloyd George is Liberal Prime Minister in UK | | https://www.gov.uk/government/history/past-prime-ministers |
| *1921* | *Study participants born* | *Study participants born* | |
| 1926 | General strike. Male unemployment high in North East of England (45% compared to national average on 14%) | 5 years old | https://englandsnortheast.co.uk/ |
| 1931 | National all-party Government formed in response to global economic depression | 10 years old | https://englandsnortheast.co.uk/ |
| 1935 | For people born in 1921 the standard school leaving age was 14 years | 14 years old | https://www.parliament.uk/about/living-heritage/transformingsociety/livinglearning/school/overview/1914-39/) |
| 1937 | King George VI crowned in UK | 16 years old | (https://englandsnortheast.co.uk/) |
| 1937 | Legislation equalised grounds for divorce between the sexes | 16 years old | https://www.thebritishacademy.ac.uk/documents/249/Happy-families-History-family-policy.pdf |
| 1939–1945 | World War 2 resulted in 384,000 soldiers being killed in combat and a civilian death toll of 70,000 | 18–24 years old | https://www.parliament.uk/business/publications/research/olympic-britain/crime-and-defence/the-fallen/#:~:text=In%20WWII%20there%20were%20384%2C000,half%20of%20them%20in%20London.) |
| 1940–45 | Winston Churchill is Conservative Prime Minister of UK | 19–24 years old | https://www.gov.uk/government/history/past-prime-ministers) |
| 1945–1951 | Clement Attlee is labour Prime Minister of UK. This government delivered sweeping welfare reforms | 24–30 years old | https://www.gov.uk/government/history/past-prime-ministers |
| 1948 | The National Health Service (a universal health system available to all the population and funded from taxation) was introduced in the UK<br>The basic State Pension was also introduced as part of the wider welfare state | 27 years | https://wellcomecollection.org/articles/WyjHUicAACvGnmJI) https://www.researchgate.net/publication/228426465_The_History_of_State_Pensions_in_the_UK_1948_to_2010 |
| 1953 | Queen Elizabeth II crowned in UK | 32 years old | https://englandsnortheast.co.uk/ |
| 1962 | First Beatles single released | 41 years old | https://www.rollingstone.com/music |
| 1969 | NASA first moon landing | 48 years old | https://www.nasa.gov/mission_pages/apollo/apollo11.html |
| 1976 | The Domestic Violence and Matrimonial Proceedings Act empowered county courts to grant orders forbidding molestation of a spouse or child. The Act was extended in 1978 to offer greater protection to women | 55 years old | https://www.thebritishacademy.ac.uk/documents/249/Happy-families-History-family-policy.pdf |
| 1979 | First female Prime Minister of UK, Margaret Thatcher, elected | 58 years old | https://englandsnortheast.co.uk/) |
| 1981 | Women born in 1921 entitled to the basic state pension and retirement from paid employment | 60 years old | https://www.researchgate.net/publication/228426465_The_History_of_State_Pensions_in_the_UK_1948_to_2010 |
| 1986 | Men born in 1921 entitled to the basic state pension and retirement from paid employment | 65 years old | https://www.researchgate.net/publication/228426465_The_History_of_State_Pensions_in_the_UK_1948_to_2010 |
| 1997 | Tony Blair becomes Labour Prime Minister in UK | 76 years old | https://englandsnortheast.co.uk/ |
| 2010 | David Cameron becomes Conservative Prime Minister in a coalition government | 89 years old | https://englandsnortheast.co.uk/) |
| 2016 | The majority of voters in the UK vote to leave the European union | 95 years old | https://englandsnortheast.co.uk/) |
| *2018–2020* | *Study participants interviewed* | *97–99 years old* | |
| 2020- | Covid-19 global pandemic | 99 years old | https://englandsnortheast.co.uk/ |

## Data collection

Semi-structured interviews explored participants' day-to-day experiences including their health, social participation and thoughts about the future. They were conducted using a topic guide which was intended to explore participants' day to day experiences, to capture their feelings about their current circumstances and set within the context of their individual biographies. The guide allowed fluidity of ideas and encouraged participants to talk freely and tell their own stories. Interviews were conducted between 2018–2020. Three participants were interviewed twice, in 2018 and then in 2020. It was intended that a second round of interviews would be conducted with all participants, but this was prevented by the Covid-19 pandemic. While one interview was conducted by telephone, due to the pandemic, it became clear that telephone and online interviews were not practical for this group of participants. All other interviews were conducted in the participants' own home and lasted approximately one hour. Six participants had a family member (daughter) present during the interview including two who lived with their daughters. One participant shared their house with their son, who was not present during the interview. Family members contributed to the interviews in practical ways, such as helping with hearing difficulties, filling in details which participants had difficulty remembering and sometimes adding their own view of the participants' stories. Interviews were audio-recorded and transcribed verbatim. Each participant was allocated a pseudonym.

## Data analysis

Data were analysed through a Positioning Theory lens in order to understand how individuals used the stories they told to construct and present their identities. Data were analysed systematically by HA. Interviews were initially coded using three-level analysis underpinned by Bamberg [22] (Table 2) and managed using NVivo 12. That is, they were categorised into the levels of self-positioning of the narrator: how the individual positions themselves to others within the narrative (level 1); how the individual positions him/herself in relation to the audience (level 2) and how they position themselves in terms of past events and pre-existing master-narratives (level 3). From this initial coding, themes were developed across the levels, taking into account the dynamic interaction between levels and reflecting the complexity of reality and lived experience. Findings are presented according to this higher order thematic analysis, allowing a greater depth of understanding about how these individuals achieve a sense of self and identity, and gain a sense of how, and in what ways, they construct who they are and mark out a place in the world. The themes were developed through familiarisation with the data and cross-cutting codes were developed into themes and then contextualised and interpreted [26, 27].

During analysis, meetings were held between the primary analyst (HA), the senior qualitative researcher (JA) and other members of the qualitative research team to discuss the themes

**Table 2. Three level positioning analysis adapted from Bamberg (2004).**

| Level | Self-positioning of narrator | Analytical insights |
|---|---|---|
| Level 1 | as a character in relation to other characters in the narrative | how this relates to actions, motivation for actions and character tropes |
| Level 2 | in relation to the audience they are telling their story to | reflects culturally embedded identity narratives |
| Level 3 | in relation to past events and pre-existing master-narratives | how the narrative fits with wider societal and cultural discourses how, and in what ways, individuals construct 'who they are' |

generated from the fieldwork material. Consensus was achieved through discussion with the team and findings were also discussed within the wider research team.

## Reflexivity

Initial interview data were collected by an experienced health care researcher who is also a registered nurse. Later interviews (2020) were conducted by two experienced health care researchers, of which one is also a palliative care doctor. Analysis was primarily conducted by HA, a qualitative health science researcher and registered nurse who has previously conducted research underpinned by identity theories. Analysis and drafts of the paper were discussed with the wider research team made up of researchers and clinicians. Specifically JA, RS and SR provided input into analysis through discussion at regular team meetings.

## Findings

Five themes were generated from the data and underpinned by the 3 levels of Positioning Theory. These related to commonalities in identity work and presentation amongst the very old: A contented life; reframing independence; familial positioning; appearance and physical wellbeing; reframing ill health. It was clear that participants utilised storylines to create a sense of self, to present their sense of identity to the audience (both researcher and wider audience) and relate these to past life events, but also to pre-existing master-narratives which framed their view of self. However, some master-narratives had been challenged by life circumstances and changes in position. It was also clear that sometimes master-narratives on which individuals could usefully draw were lacking or absent, which led to new ways of creating and thinking about identity in relation to how life was currently lived. For example, culturally accepted notions of parent/child relationships, and the associated rights and responsibilities, had often shifted leaving individuals unclear about the cultural norms available to draw on. While the world of the very old had often shrunk, expectations were altered and managed in order for individuals to continue to make sense of their place in the world and steps were taken to exert forms of control over new ways of living. This will be explored across each theme in turn.

## A contented life

A dominant theme amongst participants was that they had had, and continued to live, an enjoyable life and were content in what they had achieved and the life they lived now, *"Lovely. I enjoy every day" (Eileen)*. Participants compared their lives to those of others to position themselves as fortunate. They used past examples of difficult circumstances or hard lives to show their own life as being much better and compared themselves to peers who were not in such a fortunate position as themselves.

> *It's wonderful really that I can do all these things, when I look around and see people much younger than me and so forth. But this is life, isn't it?*
>
> *(Jack)*

Participants often drew on their own previous achievements, whether in work, hobbies or family life to illustrate past successes and status which informed their current perceptions of self. For some, past achievements were considered to afford them continued status and a well-earned position in the world. That they had provided for themselves or had been well provided for was a source of achievement.

*I'm the oldest lady member of the golf club. I'm an honorary member. At the annual dos [functions], I'm sitting at the top table with the captain. Very often, I'm presented with the flowers there. . ..It's really a wonderful life. I've had a great life, but I've worked hard. . .I've been busy, but it hasn't done me any harm. You're on the go the whole time. I've really been very lucky in what I've spent my life on.*

*(Jean)*

While for most, the sphere of present life experiences had shrunk, participants still focused on smaller life pleasures to which to look forward to. For some, expectations were self-managed and pleasure gained by focusing in the here-and-now and what is achievable, rather than what had been lost. Accepting the perceived realities of getting older, and framing expectations within this context, was considered to support ongoing happiness.

*As you get older your senses get older, as well, and they deteriorate. I accept that and I think because I accept it, I'm still a pretty happy person. . ...I can do what I want to do, I enjoy video games. When the family say, "Dad, come on, we'll go [out]" I'll go out with them, that's great. I enjoy my food. What more do I want?*

*(Alan)*

However, past events and loss of place in the world were sometimes saddening and difficult to reconcile. As a consequence, gradually withdrawing from previous life experiences and realigning new ways of maintaining a place in the world was incremental.

*I was a member of [the] Golf Club for years, and most of the bods [people] that I knew over there and played golf with have died. . ...I go occasionally, but you see, I'm a yesterday's man. . .I was captain of the golf club in 1972, but when I go over there now, instead of the usual faces that would greet me, they're all strangers . . ..I joined the bowls club. . .I felt that when I was delivering the bowl, I was just about ready to fall over, so I packed up [stopped] bowling. . ...I still go to the bowls club. I get the bus into the village and walk up the road to the bowls club. I sit there with a cup of tea and watch them play. It gets me out and sitting in the fresh air, you know, it's a change from being in the house.*

*(Bob)*

Furthermore, social circles within participants' own age group had also diminished and while some maintained friendships, to a limited extent, even that was difficult and many participants' spouses and contemporary friends and peers had died.

## Reframing independence

Independence and autonomy were presented as being central to how participants constructed and maintained a sense of self. This was narrated in a number of ways: through demonstrating the importance of controlling what was controllable, by comparing self to others and through calculated risk taking. Participants mostly presented themselves as independent and autonomous. It was considered a responsibility to maintain as much independence as possible within shrinking parameters.

*I go up for [medication] to the chemist. They could deliver it but I said, "Well, I never know when I'm gonna be in." So I'd rather go up for them myself. So I get them on a Friday, when I get some ham and some rolls*

*(Mary)*

Often retaining independence was presented as a battle between family and carers and the individual's autonomous decision making. "*She's going to go to town with me, not that I need it. I don't need help or anything, but they won't let a person my age wander around. They're very strict about that*" (Anne). As a consequence, participants picked their battles. Tensions were countered in a number of ways, however, they were rarely challenged overtly. Instead participants used what might be described as subversive tactics to maintain a level of independence. For example, humour or secrecy was used.

*My son will say to me, 'Mother, you want to take a couple of paracetamols.' Well why if I don't need them should I take them?. . ..I just ignore him as well.*

*(Judith)*

Because participants were often required to become more reliant on others, they had to compromise their previous expectations and standards. It was clear that incremental losses in autonomy occurred over time, and this was countered by articulating the gap between how life was lived currently and longstanding personal values and expectations. However, while reliance on others and acceptance of differing standards was sometimes considered frustrating, steps were taken to regain some element of control and exercise 'resistance' within a potentially unequal power dynamic.

*I had one [continence pad] lying under me and when that gets too wet, I can lift [up]. . ..One lass [person] came from the office and I said, "I think it's disgusting." She said, "Well, that's all we can do, because you're not supposed to get out of bed on your own." And that's when I waited for [carer] coming. I was sitting on the end, holding there. You know, I thought, "I'll let you see"*

*(Pauline)*

However, perceptions of independence and autonomy were relative. Despite living in assisted housing, having carer support twice daily and having meals delivered, one participant considered themselves independent. This was important as independence was seen as a proxy for dignity, which was also seen as central to concepts of self.

*I'm quite independent. . ...I get up and I make my own bed, get dressed. I don't do bad for my age. So I've still got my independence. Which is good, because along with independence comes dignity.*

*(Alan)*

Participants often compared themselves to peers who were more dependent and less able to exercise personal autonomy, *'It's awful when you're with people because the majority of people in their 90s are away with the mixer [confused]'* (Anne). By positively comparing themselves to their wider peer group, a positive self-image was created, and personal satisfaction was taken from challenging perceptions of what it is to be very old. This was particularly significant

when others, such as health care professionals, had positioned the participant as 'different' to the wider peer group, which strengthened perceptions of autonomy and independence.

> *I shouldn't be boasting but [the carers] tell me that they wished everybody was like me as far as caring for them was concerned because I do so many things for myself and because I'm not always ringing the buzzer for a carer to come. . ..Just been that way all my life really—Being independent. People say I'm too independent.*
>
> *(Gwen)*

Furthermore, storylines of bravery and risk-taking, both experienced and hypothetical, were used to demonstrate independence and illustrate a sense of control.

> *I did have an intruder once. . .I was sitting on that chair one day. I was reading. I never heard a sound. He just crept around. I jumped out of my skin, I just looked at him and said, "What do you want?" He said, "I've just been seeing your neighbour, I just wondered how you are, she said you weren't very well." I said, "I am very well thank you." He walked over and he sat on that chair. I noticed him pulling the curtains back. I could feel his hand going to grab my bag behind there. . .I was shouting. . .. "Get out of here. Get out of here this minute." I picked my stick up, "I'll hit you with a stick if you don't move." He ran out.*
>
> *(Adele)*

Indeed, risk-taking narratives were utilised in a number of ways. For some, keeping safe was considered an important responsibility, "*I wouldn't go out on my own now down to the shops in case I fall, because if I break my limbs, where would I be?" (Angela)*. For others, what might be seen as risky behaviours were considered rational and sensible workarounds to address the different safety needs of the very old.

> *I'm safe in here. . ..Since August, I've never shut my front door when I go to bed. I've left it open so that, if they [the carers] need it, they can get straight in. If you lock your door, you can't get to the door, they've got to go back to the office and get the key. By the time they're back to the office, they've got to come up. A lot can happen in a few minutes. So now I just think, "I'll leave the door open."*
>
> *(Penny)*

## Familial positioning and lack of master-narratives for new family positions

Relationships with families were perhaps some of the most difficult to navigate in terms of identity and self-concept. This centred on negotiating boundaries and changing power dynamics within family life. After years of taking responsibility for other family members, responsibility had either been actively taken on by family members, or participants had devolved responsibility to others. For some participants, this happened swiftly in response to a significant event, while for others this occurred more incrementally over time and in response to smaller events or general deterioration in health and wellbeing. While for some participants this was a relatively innocuous transition, for others, these changes in relationships dynamics were more challenging. Participants often referred to 'being allowed' or 'given permission' to undertake everyday activities of living, indicating underlying frustration, *"Yes, that's what my son said to me once. He said I was too independent. (Laughter)" (Angela)*.

In some cases this led to dissonance between perceptions of risk and maintaining autonomy. For example, after a fall one participant regretted informing his children of this, but did not articulate why. He had clearly worked through immediate strategies to help himself which he appeared to find acceptable, but his children felt that further safety measures were necessary. In this way, traditional child-parent boundaries and role definitions have shifted, however, there are no blueprints or master-narratives to which either party can relate or base their actions on. From a Positioning Theory perspective, lack of master-narratives on which to draw, and the creation of alternative master-narratives of the adult-child and ageing parent lead to incongruence in previously stable and well-defined relationships and identities.

> *I fell on the floor, and I couldn't get up. . .I shuffled on my bottom into the bedroom. I managed to get there. Then I was able to lift myself up onto the bed. Then I was alright. Well, I happened to mention it to my sons, and of course that started the ball rolling [to get call buttons/alarms]. I suppose I shouldn't have mentioned it in some respects but never mind.*
>
> *(Jack)*

For many, family members particularly daughters and, to some extent, sons, nephews/nieces and grandchildren had taken on a range of responsibilities including meal preparation, cleaning, financial responsibilities, personal care, shopping, gardening, fixing boilers, chiropody, day trips and providing social support. These responsibilities sometimes entailed significant travel and time commitments. This was despite family members often having health and ageing issues themselves, *"[son-in-law] has to come and push me now because [daughter's] hands are getting bad with the arthritis" (Mary).* Caring roles were often framed as something their adult children wanted to do to help their parent, without realising that in many instances this was a necessity. By framing the relationship as reciprocal or voluntary, it perhaps made such shifts in familial roles easier to accept.

> Angela: *Well, if they want to help, I just let them.*
>
> Daughter: *We wouldn't want to push it onto you, but we never got to the bottom of why she just stopped looking after herself. . .She didn't look after herself and didn't eat. When I came this day, she hadn't taken her tablets and she hadn't been eating.*

Similarly, when participants themselves realised they needed help and were active in that decision-making, support from families became more acceptable, although this remained difficult to concede. However, drawing on Positioning Theory, despite individuals challenging and shifting established narratives, it remained that changed societal roles would sometimes create tensions.

> Pamela: *When they got the wheelchair I thought, "I don't want this." But I had to give in in the end.*
>
> Daughter: *Oh, it was a tough battle. (Laughter)*

While some participants recognised that caring for them was in some cases a heavy burden, *"I just feel I am preventing [daughter] from doing lots of things that she would do" (Tony),* this was not always the case and there was often an assumption that caring responsibilities would automatically be transferred to children. When some children were considered less keen to take on such socially accepted roles, this was perceived negatively and deviated from socio-cultural narratives of what it is to be a 'good' son/daughter.

*The son, I didn't see very much of him. [Daughter] looked after her dad as a carer and looked after me. Once her dad died, they automatically put her over as my carer. That's how she seems to do a lot, because she's the carer.*

*(Penny)*

Some participants considered it was important to plan for the future, but few had taken steps to actively and practically address or make plans for end-of-life care or consider what would happen if their health deteriorated, "*No, we'll cross that bridge when we come to it.*" *(Alan)*. It appeared that while funeral plans were more common, the processes of dying were not considered or planned for. Instead focus appeared to be more on what might happen after death occurred.

*I've got my funeral all paid for.. .. You can't do anything about it [dying] can you? And I've had a good life*

*(Pauline)*

However, while practical end of life plans were not commonly made, this did not mean that participants had not considered future effects on self and their family. Instead, different sorts of planning for the future were carried out, for example praying for a comfortable death, which was associated with dignity and peace. In this way, individuals could make sense of the inevitability of dying by drawing on established religious and cultural understandings of what this might mean for both the individual and their family and by centring dignity and autonomy, both of which were considered important identity constructs.

*It's not only for myself that I'm concerned, but I'm concerned about the people around me. . .. I'd rather just go quietly with dignity. I think he [God] might answer my prayers, because I ask him often enough.*

*(Alan)*

Perhaps inevitably, the greatest challenge to familial roles and identity was the loss of a child. With few past experiences or societal cues to draw on, this was particularly difficult to deal with. That accepted social order had been disrupted made the reality difficult to comprehend. Nonetheless, this was accepted stoically and, reflecting the view [21] that master-narratives are not fixed and are open to challenge by individuals who have agency and are active in their identity construction, an element of acceptance and sense-making could be developed.

*I had an awful experience. My older son died in July. It was hard. It's the worst thing that's happened. You don't expect your family [to die]. . .It was hard to watch him deteriorate. It was a year yesterday and I went up with a few flowers, but I'm not so bad now. I've got over the worst. . ..I often think of him*

*(Maureen)*

Despite some tensions and challenges in negotiating changing familial relationships, it remained that for those participants with children and close relatives, these relationships were of central importance and companionable and close relationships remained. Joe says about his daughter, "*She's a diamond, she really is a diamond*".

## Appearance and physical well-being

While a minority of participants directly referred to issues around appearance, for many, underlying narratives implicated appearance and physical wellbeing as a barometer for standards, demonstrating the outward 'face' of capability and living well. The majority of female participants mentioned either weekly trips to the hairdressers, or the hairdresser coming to their home, which was considered an outward presentation to others of self-worth, *"I think if your hair is a mess, you're all a mess. If you go out some place and your hair is untidy—That's one thing, I always look after my hair" (Jean)*. For those who raised the issue, appearance was considered an important part of identity and was related to maintaining standards and looking after yourself, *"I'll have to get really smartened up for the summer" (Anne)*. Being outwardly presentable was considered analogous with ability to function and demonstrated capability. Feeling and looking younger than one's age was considered to be positive affirmation, with others considering looking younger to be complementary. This created a sense of pride and added to a positive sense of self. In relation to Positioning Theory, this outward positive presentation of self to an audience shores up self-image on a world where, particularly for women, ageing is negatively perceived.

> *She brought this lady doctor in, she said, "If I get to 95, I hope I look like you." I always put a little bit of make-up on. Not a lot. Just a little bit. Tinted moisturiser and that. I don't know. I think it is maybe my upbringing.*
>
> *(Maureen)*

By presenting taking care of oneself as a lifelong characteristic, the maintenance of a previously established aspect of identity was preserved. Indeed physical appearance continued to be seen, by some, to be important after death, *"[when making funeral arrangements] I said, 'You'll make me nice, won't you?' She [the undertaker] said, 'Pauline, I'll do you lovely'." (Pauline)*

Appearance also played a part in memories of past, and sometimes happier, times. There was a sense of loss for the life events and interests which were now unable to be pursued. *"I still look at fashions on the telly, for all I couldn't wear them if I had them, but I still think that way you know." (Maureen)*. Referring to past experiences provided a thread to current perceptions of self and how this was conveyed to others. In this way, past experiences and master-narratives were used as a mechanism to connect to others in the present.

> *I was 85 when I got my last [trophy for dancing]. Yes. Latin American. . ..Oh, I had some beautiful [dresses]. There's one [photo] I've got of him and me and [the dress is] a turquoise blue, and I've still got my little white fur coat. When I tell them [the carers] to look for something, they say, "God, strewth, why all the coats and the dresses?" I say, "Oh, I love my clothes." I did then. But never mind.*
>
> *(Pauline)*

## Reframing ill health

Participants appeared reluctant to acknowledge progressive ill health and could be seen to minimise quite serious events. Often individuals would present themselves in their narrative as being able to cope with even significant health issues. *"I fractured my hip. That was three weeks [in hospital]. . ..Yes, it was alright. I soon got over that" (Adele)*. It seemed to be important that ill health or particular events were seen as manageable and not a reflection of diminished

independence and autonomy. Claims of ill health could be seen as a threat to autonomy. Consequently, positioning self as healthy, capable and safe was common among participants. This was potentially to protect against as the associated risks to autonomy posed by loss of health status.

*When I had this last fall, I told the doctor, "I don't fall." He said, "Well, what do you call it?". I said, "A slide". I don't even hurt myself. I just slide down…..It was funny, the last fall I had …..The doctor up there said, "Well, we're trying to put you in a home" I said, "Don't you dare".*

*(Pauline)*

In many cases, multiple and complex health conditions had been minimised to the point of participants not recognising the extent that health problems impacted on their lives. Work-arounds had been developed to reduce the impact of limitations, but it remained that life could be considered to be severely limited. However, this had been normalised and, in a large part, accepted. This is exemplified in the following truncated quotation taken from a longer narrative discussion, where it was only probing from the researcher which elicited the extent and multitude of health conditions which require daily negotiation, management and compromise.

Penny: *I don't feel anything wrong. Probably the arthritis, that's the only thing. I'm riddled with arthritis. My hands, yes. When I say arthritis, I haven't got arthritis fingers.*

Interviewer: *So it doesn't really stop you from doing [what] you want to do?*

Penny: *It stops me from standing. I could stand, make a cup of tea. I might put the water in, put the teabag in, come and sit down for a bit while it's mashing [brewing] and then go back. I can't stand all that time until the tea is ready.*

Interviewer: *So you find ways around?*

Penny: *Oh, yes, I find a way around some way.*

Interviewer: *Are there any other health problems that affect you day-to-day?*

Penny: *Just my arthritis—apart from taking those tablets. That's all.*

Interviewer: *Yes. You're sitting here with your legs up?*

Penny: *Yes, because I've been told to sit with my legs up. The point is they were filling with water. Whenever a nurse comes, they say, "Are you keeping your legs up? That's why I sit this way, it's easier.*

Interviewer: *Do you see a nurse quite often?*

Penny: *Every time I hurt myself.*

Interviewer: *Yes. I see you've got a dressing on your arm there, quite a big one it looks like?*

Penny: *That's just the size of the plaster. I knocked it on [the walker]. I broke the skin. I've got very fine skin. See how I bruise. Yes, they've got me on warfarin. That's how I've got all these marks on me now. Oh yes. I can feel a pain there. The least little knock and…then you get the sore. Whenever it bleeds, you have to get a nurse in. They're never away.*

Some participants considered adverse health events, and waning health more generally, to be a source of embarrassment, particularly when they occurred in public. Consequently, framing ill health through minimising symptoms and outcomes appeared to be a strategy many participants shared. In relation to Positioning Theory, participants presented themselves to the audience (friends, health care professionals, family, researchers) as both relatively healthy and able to deal with any health concerns they did have. Most participants did not want to be seen to make a fuss or draw undue attention to themselves or their situation, or appear to be unable to cope in any way.

*I was so embarrassed and flustered that I just sat quietly. . .. show[ing] myself up like that. Eventually word got through the ladies. "Susan has passed out. [She's] had a bad turn". . .They all got up and came to say, "Eeh what's the matter" and I'm sitting and I just felt–I'm sitting stupid looking and–just embarrassed. I just felt that everybody's standing looking at you and it was at the back of my mind, "Do they think I've had too much to drink?" and I mean I hadn't*

*(Susan).*

Related to this is the lack of control experienced by some participants, where others took over responsibility during ill health events leading to reduced autonomy and a sense of frustration.

*I was down in the shop and I just felt swimmy headed [light-headed]. I didn't faint and the girl in the shop sent for the ambulance, [the paramedics] took us to the [hospital] and I thought, "What are they taking me here for?" I kept saying, "There's nothing wrong with us", but nobody listened and they took us there.*

*(Angela)*

While this was not directly linked by participants, there was an element of risk taking and subversion of what might be considered expected behaviour. For example, some participants did not routinely use walking aids or emergency call systems which could be considered to increase their risk of adverse health events. There appeared to be a dissonance between being aware that such practical measures were necessary, but a reluctance to acquiesce to delegating decision-making and control to others. Personal call systems (buttons/buzzers), which would alert a central system to send emergency help when activated by users, appeared to be a particular issue with multiple participants resisting their use.

*"I don't wear [call button] no. No, I haven't got anything like that. I seem to remember having one at one time. Maybe I have put it away. . . (Laughter)"*

*(Margaret).*

Some participants deviated from formal safety measures as they perceived that others would think they could not cope independently at home and, as a consequence, autonomy would be reduced further. Consequently, what might be considered puzzling behaviour in terms of maintaining safety, can also be understood as a way of maintaining control over decision-making.

Pauline: *the last time I had the fall where I slid, I was on the floor and I shouted [neighbour] and she came up.*

Interviewer: *Do you not use your [call] button*?

Pauline: *Well, when I got up I wouldn't send for them [the emergency responders via the call button]. [Neighbour] put me to bed. . ..she asked [another neighbour] to come and help her to pick me up*

Interviewer: *So, your two neighbours helped you. You didn't end up using your call service*?

Pauline: *No.. ..I was vexed with one of my granddaughters because she said, "You know, Grandma, would you not be better in a home." (Gasps) Oh, dear me. If I could have got up!. . . I said, "No, I wouldn't be better in a home, I like my home."*

Related to this was the rejection of safety and support aids more broadly, mainly because participants did not view themselves as in need of them, through which was derived a sense of pride. In terms of Positioning Theory, this outward presentation of capability is a tool to distance self from societal expectations of what it is to be very old. *"People are surprised that I don't need support. I go up the stairs and down. It's like climbing Mount Everest going up, (Laughter) but I don't want a stairlift at all" (Margaret).* However, the rejection of safety and support aids exposed participants to risk. As a consequence maintaining safety and expressing autonomy was a finely balanced risk calculation and different for each individual.

Ill health was often reframed as occurrences which were an expected and *normal* part of being the oldest old, "*It's just macular degeneration, it's not that I have a problem (Anne).* This normalisation was, in part, achieved by positioning themselves in comparison with their peer group. It was presented as *expected* and *normal* for people in this age group to experience specific issues, and in this way these shared issues were normalised. A sense of inevitability led to acceptance by most that deterioration in health was a consequence of ageing and to fight this would be futile.

*They told me they'd found [cancer]. . .. I take the bad with the good. . .if my health deteriorates, I have no outlet. I've no avenue to support. None at all. I'm in the lap of the Gods.*

*(Malcolm)*

Acceptance also meant that worrying about health issues and even the inescapability of dying was reduced to a manageable level. This acceptance appeared to coincide with less interaction with healthcare services.

*I don't think I'm seriously concerned about my health. I just take it as it comes and I reckon sooner or later I will fall ill and that will be it. And okay, I'm not worried about that*

*(Russell).*

By drawing on Positioning Theory, several themes are identified which are central to the ways in which this group of very old people maintain and reformulate identity in a world that can pose significant challenges. People in this study present themselves as being content with life, both currently and as lived. By positioning themselves relative to other characters within their narratives, by presenting self to their audience in specific ways and by drawing on, modifying, re-creating and sometimes refuting socio-cultural master-narratives, identity work can be seen to be a lifelong pursuit of sense-making through reframing narratives to fit shifts and challenges to concepts of independence, health, family roles and appearance and physical wellbeing.

## Discussion

By exploring the everyday stories people tell about their lives through a Positioning Theory lens, we were able to shed light on how this group of very old individuals aged over 95 years modified and created an ongoing identity and sense of self. This group receives little attention and such studies are rare both in the UK and internationally. Exploring identity work in this distinct group of individuals is important as, despite an increase in the number of people in the UK over the age of 90 years, this remains a very select group, with potentially distinct and different needs, values and ways of living their lives. Based on 2021 UK Census data [28], there are 609,000 individuals >90 years old living in the UK, only 20% of which are aged between 95–99 years old and 2.5% are ≥100 years. There are more women than men aged ≥95 years with 2.8 females to every male in the 95–99 year age group and 4.6 females to every male for people aged ≥100 years and over, in the UK in 2020, although this gap is narrowing. Consequently, this paper contributes to our understanding of what it is to be very old in contemporary society.

By analysing data in this way we were able to see how individuals created a sense of who they are now, by how they talk about and present themselves to self and others, and how they draw on, challenge, refute, adapt and create established societal master-narratives to explore and explain their place in the word. While individuals narrated identity in different ways, there were commonalities in these stories and ideas of self-concept. Participants in the study saw themselves as largely content and happy in their lives. Despite their world becoming smaller, and health and other challenges, participants were able to find pleasure and contentment in small daily routines and events. Past relationships, events and experiences were drawn on in order to make sense of ongoing ways of living. While there were some frustrations and tensions around feelings of loss of autonomy and independence, small acts of resistance and subversion were acted out in order to maintain some sense of control. Perceptions of self and identity were reframed in order to maintain a sense of autonomy within more narrowly defined parameters. Participants appeared reluctant to acknowledge progressive ill health and could be seen to minimise quite serious events.

In our study, participants presented themselves as largely independent and young for their age. They perceived themselves to not be suffering from ill health and had reduced interaction with services such as health services, unless specifically required or in case of emergencies. In a study about seeking assistance in later life [29], people aged 68–95 did not perceive they needed assistance, despite evidence to the contrary. They maintained independence by gradually modifying their lives and were keen not to rely on the help of others, with unsolicited help often considered unwelcome. Instead, they engaged in self-management and reframed ill health as 'old age'. This meant that older people were not receiving the care they required [29]. However, there is often conflict between healthcare professionals' views that reducing physical risk is the priority, while losing identity, autonomy and dignity is the main issue for those potentially in receipt of that care [30]. That participants did not perceive themselves of being in need and modified their lives to fit current positive perceptions of self are reflected in the findings of our study, where participants characterised their lives in a positive way and valued identity and autonomy. For the participants in our study, it appeared that maintaining these important aspects of self was central to enjoyment of life as it is now.

Running through the narratives in our study was the central tension between an understanding of health as the absence of disease, as characterised by Boorse [31] and adopted by some professionals and family members referred to in this study, and more positive models of health, such as Nordenfelt [32] who focuses on ability and a conception of health beyond the absence of disease. Similarly, Sen's capability framework [33] suggests that it is morally

imperative for individuals to achieve the sort of life they find valuable, based on their own vital goals and wishes, while Huber and colleagues [34] see health as the ability to self-manage and adapt to change. In a Dutch study of individuals aged over 85 years [35], successful aging was perceived as a process of acceptance and adaptation. It is these more positive conceptualisations of health that appear to have been adopted by the very old participants in this study. However, such conceptualisations may mask the extent of assistance required, with burden often falling on family and concealing the need for formal support [29]. As our study was based on self-reported narratives, the extent to which masking, or missing out on formal support, was problematic is unclear. What was clear was that participants negotiated and relied on informal support.

In our study, family relationships had become increasingly complex and in some cases imbued with ambivalence, with societal roles challenged by changing dependency and lack of a blueprint, or master-narrative, guiding how such relationships should be played out in the contemporary world. This was particularly relevant for those whose adult children were often experiencing their own challenges associated with health and ageing. The burden on children who would also be considered to be older people is unclear, but a study of family caregivers for older people in Ireland found a lack of support for family carers [36]. 30 percent of carers were themselves over 64 years of age, with 28 percent reporting mild to moderate carer burden and a further 36 percent reporting moderate to severe carer burden. In addition, 78 percent of family careers had physical health problems and 42 percent reported mental health issues [36]. That adult children are also experiencing life as an older person means that societal roles are both challenged and unclear. Furthermore, family dynamics have been found to change over the course of moving from being 'young-old' to 'old-old', with the 'younger old' often providing support to their adult children, for example, with childcare [37]. These responsibilities then shift as it is the parent who then requires support from the adult child. This shift in division of responsibilities has the potential to destabilise sense of self and requires complex renegotiation of relationships.

Participants in our study negotiated changing relationships with caregivers and family members in a number of ways. Despite delegating specific roles to others, including family members, and acquiescing to some loss of power, participants worked to maintain their autonomy in ways which were acceptable to them. This often meant strategically deciding which battles to 'fight', ignoring or acting counter to advice and guidance from their children, other relatives and carers and actively deciding not to worry about some aspects of their lives that would previously be important to them. In these ways a sense of autonomous self was maintained, despite ongoing and continued challenges to independence. Reliance on family members, while appreciated and to some extent expected, was flipped to allow the belief that carers chose to and wanted to help, or that the relationship had reciprocal benefits, rather than this being seen as a necessary support.

While some relationships with same age peers persisted for participants in our study, this was quite severely limited as many peers had died. As a consequence, such relationships and identity work were not particularly visible in the narratives of participants. It may be that these relationships had become less significant as same age peer groups had shrunk dramatically.

It was clear that, from a Positioning Theory perspective, individuals in this study renegotiated and reformed relationships with significant others through presenting themselves to, and interacting with, their perceived audiences in ways in which maintained their sense of self. It was also apparent that the perceived audience also influenced expectations of behaviour. This presentation of self is reminiscent of the work of Goffman [16]. For Goffman, people perform social interactions through which they manage the impressions given of themselves to others. This relies on all parties involved working with an underpinning shared understanding and

social scripts guiding how self and others should behave. However, our study identifies diffi-culties can occur when there is a lack of shared understanding underpinning interactions. As a consequence, individuals are required to negotiate relationships without the usual social guid-ance on which to draw.

A positive sense of self was also maintained by taking pride in appearance and well-being, being considered mentally and physically young in comparison to chronological age and through positive comparison to peers. This has some resonance with downward comparison theory [38] where individuals positively compare themselves to others as a mechanism to reduce the impact of distress or negative life events. Positive comparison to peers can be seen as a tactic to distance individuals from shared stigmatised perceptions of old age [9]. While individuals may refute negative ageist stereotypical master-narratives in relation to their own perception of self, the same individuals apply these negative stereotypes to others in order to distance themselves from being old or ill [9] and this was reflected in our study. Our study is also consistent with previous studies which found having an age identity which is younger than chronological age is associated with increased life satisfaction [39, 40]. Because in western cultures youth is positively associated with physical wellbeing and mental acuity, and ageing is viewed more negatively, having a younger identity is considered central to the outward display of competence [39]. Furthermore, the impact of a more youthful age identity differs along gen-der lines, with females in middle age more likely to be judged according to traits associated with youthfulness and physical appearance [41]. While the men in our study did not discuss appearance, women associated concepts of dressing smartly, styling hair and being considered by themselves and others to be younger than their chronological age to be a positive indication of mental and physical wellbeing. Furthermore, they drew on past recollections and events and related this to a continued positive sense of self. In this way, long held identity narratives threaded through and informed current perceptions of self. It was of note that appearance was the only aspect of the narratives that appeared to differ along gender lines, with other themes, and the ways in which identity was expressed, being quite consistent between men and women in the study. Given the study's relatively small sample size, and the even smaller subgroup of male participants (30% of the 23 participants were male), it may be difficult to draw inferences from this. Furthermore, not all females in the study referred to appearance, However, it remains that appearance was important to the expression of identity in some female partici-pants in this study.

Sneed and Whitbourne [13] theorise this is underpinned by a balance between identity assimilation (where individuals focus on maintaining an identity consistent with a previous sense of self concept, even when evidence challenges this) and identity accommodation (shifts in identity in response to changing experiences). While our study is consistent with such described theorised concepts of age identity, it is of note that other studies focus on a much younger age group, with 'old' being classified as >65 years, compared to our group of nonage-narians. As one participant in our study pointed out, being 'very' old poses its own challenges and there are significant differences in the experiences of those who society defines as old, and the oldest in society.

Linked to this was the reframing of ill health as manageable through minimisation, distanc-ing and positively comparing self to others. Risk of falling, for example, was conceptualised in a number of ways. Similarly, in a previous study [42] older people perceived falling to be a threat to positive self and social identity because falling was seen as stigmatising and a chal-lenge to competence and autonomy. To counter this, participants distanced themselves from negative tropes by comparing themselves positively to others as 'the type of person who does not fall'; by excusing falls as random or extrinsic events outside of their control, or attributing

falls to not taking care, rather than due to physical decline. In our study, participants used similar tactics to present themselves as different to, and more competent than, their peers.

Furthermore, our study identified that risk of adverse events, such as falls, was finely balanced against the risks of diminished autonomy. For example, some participants chose not to wear call buzzers/buttons as they appeared to consider the 'risks' of displaying themselves to be incapable, or not in control, to those who they considered to have authority over their lives outweighed the risk of becoming unwell and not being able to summon help. Fear of embarrassment and losing control have previously been identified as potential social consequences of ill health [43] and concepts of autonomy and identity contribute to maintaining dignity in older people, with the risk of loss of autonomy and identity outweighing any immediate risk to health for some [44]. Furthermore, social status may be undermined by being publicly stigmatised as old and frail, while for some, help, advice or interventions are considered unwanted and, in some cases, patronising or insulting [43]. In our study, participants could be seen to subtly, covertly and actively resist 'help' or measures which they considered to undermine or impose on their autonomy in a way which balanced their rights with the concerns of others. While there were few 'rules' about how to negotiate these difficult concepts, amicable relationships had been forged.

Normalisation of health issues and the acceptance of dying which were characterised by participants in our study were reflected in a study of ≥95-year old's attitudes to dying [4], which found death was an accepted part of life for the very old, that there had been little discussion of death or planning for end of life with families or carers, that people were not worried about death, but about the process of dying itself. They wanted to die peacefully and often put their trust in religion. However, in contrast to our study, some participants felt they wanted to die and felt their quality of life was minimal, with many expressing they were ready to die or that they had lived too long. In our study, participants largely expressed enjoyment of their current life. However, it is of note that the focus of our study was everyday life, not specifically eliciting attitudes to death and dying. Nevertheless, it remains that in interviews where participants were relatively free to talk about various different aspects of their lives, what was important to them and plans for the future, participants in our study talked positively about their lives and did not, on the whole feel they had lived too long or wanted to die.

Small acts of resistance and assertion of control commonly recurred throughout narratives in our study and the associated themes generated. Directing carers how to carry out procedures and where to put items, negotiating bedtimes, defining selves as 'non-fallers', deciding not to take medication and resisting contemplating the future were all utilised in a way which can be seen to maintain independence, autonomy and dignity in the face of diminishing control. While participants 'picked their battles' and did not outwardly object to help offered by others, these acts of resistance can be seen as a way to counter and reject the perceptions of others that participants needed support to manage their own lives and that safety should be prioritised over other aspects of life which they perhaps valued more, such as autonomy and identity.

## Strengths and limitations

This study reports the narratives of a group of very old people in society whose voices are rarely heard. As such, it was intended to present a broad-brush overview of multifaceted aspects of identity. Future research could build on this by focusing on specific areas e.g. risk or comparison with other age groups. It is of note that most of the current literature focuses on 'younger' old people, and while some common themes have been identified, the need to explore the experiences of the very old is clear. Consequently, we were interested in, and chose

to focus on, the experiences of those aged over 95 years as data about experiences of this group of individuals are lacking. While some other studies may include the very old, this study focuses solely on this group and should be viewed within this context. Therefore, this study contributes to our growing knowledge of the experiences of the very old and can be used as a baseline from which to develop further study. This study focuses on the narratives of a group of individuals who are part of a larger longitudinal cohort study and who chose to take part in qualitative interviews. Consequently, we do not claim the findings of this study are representative across different contexts, but instead offer insights which are reflected in the small but growing literature relating to the experiences of the very old and may resonate with this group more broadly.

## Conclusion

This study gives voice to the seldom heard narratives of a group of very old people in society. As such it sheds light on the experiences and views of a group of people of whom little is known about how they make sense of their world. Specifically, identity work in the very old has been given little consideration, but is important as findings both challenge common perceptions of how life is experienced in this context and provide insight into how the very old negotiate, acquiesce, resist and challenge the world around them.

## Acknowledgments

We would like to thank participants for contributing their experiences.

## Author Contributions

**Conceptualization:** Helen Anderson, Rachel Stocker, Sian Russell, Lucy Robinson, Barbara Hanratty, Louise Robinson, Joy Adamson.

**Data curation:** Rachel Stocker, Sian Russell, Louise Robinson, Joy Adamson.

**Formal analysis:** Helen Anderson, Rachel Stocker, Sian Russell, Joy Adamson.

**Funding acquisition:** Barbara Hanratty, Louise Robinson, Joy Adamson.

**Investigation:** Rachel Stocker, Sian Russell, Lucy Robinson.

**Methodology:** Helen Anderson, Rachel Stocker, Sian Russell, Lucy Robinson, Barbara Hanratty, Louise Robinson, Joy Adamson.

**Project administration:** Rachel Stocker, Sian Russell, Lucy Robinson, Louise Robinson, Joy Adamson.

**Resources:** Rachel Stocker, Sian Russell, Lucy Robinson, Joy Adamson.

**Supervision:** Joy Adamson.

**Validation:** Barbara Hanratty, Louise Robinson.

**Writing – original draft:** Helen Anderson, Joy Adamson.

**Writing – review & editing:** Helen Anderson, Rachel Stocker, Sian Russell, Lucy Robinson, Barbara Hanratty, Louise Robinson, Joy Adamson.

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
