## [Decision Letter · Decision Letter 0]

29 Jul 2022

PONE-D-22-09522Identity construction in the very old:A qualitative narrative studyPLOS ONE

Dear Dr. Anderson,

Thank you for submitting your manuscript to PLOS ONE. After careful consideration, we feel that it has merit but does not fully meet PLOS ONE’s publication criteria as it currently stands. Therefore, we invite you to submit a revised version of the manuscript that addresses the points raised during the review process.

Both reviewers have highlighted the value of this study, and agree on the fact that is well-conducted. Yet, they point to some revisions that I kindly ask the authors to address before I can make a decision about publication. 

We look forward to receiving your revised manuscript.

Kind regards,

Sara Rubinelli

Academic Editor

PLOS ONE

Journal Requirements:

a) Did participants provide their written or verbal informed consent to participate in this study?

b) If consent was verbal, please explain i) why written consent was not obtained, ii) how you documented participant consent, and iii) whether the ethics committees/IRB approved this consent procedure

Reviewers' comments:

Reviewer's Responses to Questions

**Comments to the Author**

1. Is the manuscript technically sound, and do the data support the conclusions?

Reviewer #1: Yes

Reviewer #2: Yes

2. Has the statistical analysis been performed appropriately and rigorously? 

Reviewer #1: N/A

Reviewer #2: N/A

3. Have the authors made all data underlying the findings in their manuscript fully available?

Reviewer #1: Yes

Reviewer #2: No

4. Is the manuscript presented in an intelligible fashion and written in standard English?

Reviewer #1: Yes

Reviewer #2: Yes

5. Review Comments to the Author

Reviewer #1: This article furthers our understanding of very old people construct their identities and sense of self in general and their attitudes and representations to daily life concerns and health in particular. As such it represents a novel rich contribution to the literature. Through the well-chosen lens of Positioning Theory (Bamberg) each person is assumed to build and perform their identities taking into account their past histories, present circumstances and envisageable futures while adjusting to economic, societal and cultural shifts over time. This theoretical framework also takes into account that narratives will be constructed in relation to listening audiences. Through well-conducted sensitive and interview guided qualitative interviewing with the old people themselves and carers and subsequent thematic qualitative analysis life narratives and attitudes to autonomy and health were well explored. Of particular interest were the different strategies used by the very old people themselves to preserve autonomy from well-meaining (over) protective families and carers. This highlights the importance of both having and seen to have control over one's own life and choices given the advent of progressive limitations on functioning. The authors rightly point out that ageing in our society is often viewed through a biomedical lens which tends to highlight its negative impact on health and well-being and economic circumstances and indeed tends to stigmatise elderly people going through this process.

Having well explored and revealed different attitudes to health and autonomy nevertheless I feel the authors have missed an opportunity to highlight a key tension revealed through analysis of the narratives running through the article between negatively valent models of health (Boorse, Daniels) and positively valent models of health (Nordenfelt). Thus it strikes me on the whole the very old in this smallish sample would seem to favour a definition of health based around autonomy and the achievement of one's own vital goals and wishes (cf Sen's Capabilities ) rather than the habitual view of health being based on the lack of pathology often implicitly favoured by protective carers and family members and the medical profession. I feel this excellent paper could be improved through referring to the above mentioned philosophy of health literature.

Reviewer #2: See attached file with full responses. See attached file with full responses. See attached file with full responses. See attached file with full responses. See attached file with full responses. See attached file with full responses. See attached file with full responses. See attached file with full responses. See attached file with full responses. See attached file with full responses.

6. PLOS authors have the option to publish the peer review history of their article (what does this mean?). If published, this will include your full peer review and any attached files.

Reviewer #1: **Yes: **Dr William Sherlaw

Reviewer #2: No

---

## [Author Response · Author response to Decision Letter 0]

14 Sep 2022

Reviewer 1 Comment: 

This article furthers our understanding of very old people construct their identities and sense of self in general and their attitudes and representations to daily life concerns and health in particular. As such it represents a novel rich contribution to the literature. Through the well-chosen lens of Positioning Theory (Bamberg) each person is assumed to build and perform their identities taking into account their past histories, present circumstances and envisageable futures while adjusting to economic, societal and cultural shifts over time. This theoretical framework also takes into account that narratives will be constructed in relation to listening audiences. Through well-conducted sensitive and interview guided qualitative interviewing with the old people themselves and carers and subsequent thematic qualitative analysis life narratives and attitudes to autonomy and health were well explored. Of particular interest were the different strategies used by the very old people themselves to preserve autonomy from well-meaining (over) protective families and carers. This highlights the importance of both having and seen to have control over one's own life and choices given the advent of progressive limitations on functioning. The authors rightly point out that ageing in our society is often viewed through a biomedical lens which tends to highlight its negative impact on health and well-being and economic circumstances and indeed tends to stigmatise elderly people going through this process. 

Response to reviewer 1:

Thank you for your comments, we are pleased to have a positive response to our manuscript and appreciate the time taken to provide detailed feedback, which we feel sure will strengthen the manuscript further.

Reviewer 1 comment:

Having well explored and revealed different attitudes to health and autonomy nevertheless I feel the authors have missed an opportunity to highlight a key tension revealed through analysis of the narratives running through the article between negatively valent models of health (Boorse, Daniels) and positively valent models of health (Nordenfelt). Thus it strikes me on the whole the very old in this smallish sample would seem to favour a definition of health based around autonomy and the achievement of one's own vital goals and wishes (cf Sen's Capabilities ) rather than the habitual view of health being based on the lack of pathology often implicitly favoured by protective carers and family members and the medical profession. I feel this excellent paper could be improved through referring to the above mentioned philosophy of health literature.

Response to Reviewer 1:

Thank you for your suggestion to highlight the tensions between study participants and their carers/professionals, underpinned by philosophy of health literature. We agree that this would strengthen the paper and have added a section in the discussion which highlights this. We feel this adds to the theoretical underpinning of the study findings and strengthens the discussion. (Page 37, para 1)

Reviewer 2 Comment: 

Thanks for the opportunity to review this manuscript, which I enjoyed reading and I think can make a valuable contribution to the literature. To situate my perspective as a reviewer without giving my identity away, I’m a person who studies the life course using mixed methods. My particular interest as they touch this subject is in the health and aging of returned soldiers. So I read the manuscript with somewhat of an eye to what we can learn from studying this particular group, and how that might inform how we study other groups of aging people. The valuable contribution of this manuscript is the engagement with a rarely interviewed demographic, and I think it will be of interest to people because of that. The paper makes its methodological approach clear with the use of Positioning Theory. There are a few ways in which I think the paper could, with minor modifications, be of interest to an even wider audience

Response to Reviewer 2

Thank you for your valuable feedback and comments on our paper. We are pleased that you feel it can make a valuable contribution to the literature and we are proud to share the voices of this group pf participants. We feel that the feedback provided will strengthen and further improve our paper and we appreciate the time taken to provide this detailed feedback. We have addressed your comments and suggestions below.

Reviwer 2 comments:

1) More clearly articulating general insights we can learn from the study of identity work in this oldest old group. The themes articulated by the elderly in this specific setting are likely to be echoed elsewhere, though with perhaps different inflections or emphases reflecting the social setting of aging adults in different societies. Perhaps in the discussion the authors could address social situations which may be isomorphic to the oldest old. One analogy that comes to mind is how different generations of workers in workplaces relate to each other. 

a) Related to this, the authors could be more explicit on why 95+ is distinctly different than merely being in one’s late 80s or early 90s (I looked up UK life tables as I was reading this, and it really emphasizes how selective this group is compared to the mere 86 year olds. https://www.ons.gov.uk/peoplepopulationandcommunity/birthsdeathsandmarriage s/lifeexpectancies/datasets/nationallifetablesunitedkingdomreferencetables)

Response to reviewer 2:

1) Thank you for your feedback, which we have reflected on carefully and appreciate. While the themes articulated by the participants in this setting may be echoed elsewhere, we feel it is beyond the scope of the paper to include other isomorphic social situations within this paper. This is because we want to focus on the clear narrative of the paper i.e. identity construction in the very old. We wanted to centre this group of individuals who are rarely researched, but are growing in number. We are concerned that diversifying to other situations, such as how different generations of workers relate to each others, may dilute and confuse the messaging of the paper. We feel we have clearly articulated general insights gained from the study of this group and it for further research/papers to apply this to other groups.

a) We agree that we could be more explicit about why this particular group of individuals are distinct and potentially different to other groups. We also agree that this age group is very select and it would be helpful to emphasise this in the paper. Therefore we have added text at the beginning of our discussion section to reflect these comments (page 34 paragraph 2

Reveiwer 2 comments 

2) Implicit in the manuscript as it stands is that the oldest old are doing a lot of relationship work. As currently framed, the paper emphasizes the individual work and presentation of self. I would encourage the authors to be more explicit about the common thread in these discussions of re-negotiating relationships with e.g. adult children and caregivers. I found it curious that there was no citation to, or engagement, with Erving Goffman and work that draws on his insights about the presentation of self in everyday life. I’d encourage the authors to make that connection to sociological work on identity and relationship re-formation more clearly. 

Response to reviewer 2:

2)Thank you for your feedback, which we reflected on considerably. When planning the study and analysis we considered how we would handle the data and the number of different analytical approaches we could have taken. As a team we made the decision to use a positioning theory lens, which draws on a social psychology perspective, as we felt this best supported the generation of ideas about the identity work which went on in this group of participants. We understand that there may be commonalities between Goffman’s work and Positioning Theory and the way the study is presented in this paper, and agree this is an analytical stance we could have used. However, as a team we took the decision to underpin the study using positioning theory and have been careful to try not to confuse or dilute what we feel are the important messages in this paper through integrating other theories. We have, however, taken on board the feedback and have added a section in the discussion about the relational aspects of renegotiating self (and relationships) and about how this relates to the work of Goffman, with the appropriate citation (Page 39, para 2). We feel, however, to go further than this would go beyond what we feel comfortable doing with our dataset and beyond the story we wanted to tell (Braun and Clarke, 2019). We have attempted to balance feedback with our own understanding of how the study should be framed. Some of the challenges of balancing peer review suggestions with authors’ understanding of the data are discussed in Sheard (2022)

Braun, V and Clarke, V (2019) Reflecting on reflexive thematic analysis. Qualitative Research in Sport, Exercise and Health, 11 (4) (2019), pp. 589-597

Sheard, L. (2022) Telling a story or reporting the facts? Interpretation and description in the qualitative analysis of applied health research data: A documentary analysis of peer review reports. SSM - Qualitative Research in Health, 100166, ISSN 2667-3215. https://doi.org/10.1016/j.ssmqr.2022.100166.

(https://www.sciencedirect.com/science/article/pii/S2667321522001287)

Reviwer 2 comments:

3)The specific context of the study is stated, but what it means to be 95+ in the northeast of England (obviously varies!) in the 2010s and early 2020s isn’t described. Specifically here, I’d like the authors to be clearer on the life course and cohort aspects of their sample. Particularly for readers from other cultures, or for readers from the same culture reading this later, what does it mean to have been born in the 1920s (approximately) and survived 95+ years. In broad strokes, some of the likely contours of family life, work, and gender relations that this group would have encountered will be helpful to contextualize the responses.

Response to reviewer 2:

3)Thank you - this feedback is really helpful. We agree that it would be beneficial to provide context to this particular cohort of participants. To this end we have added a timeline in order to situate the potential life experiences of participants, and an explanatory sentence in the methods section. Because the interviews did not specifically collect consistent data on aspects such as previous family life and relationships, employment, geographical location over time etc, we have been careful not to go beyond what is explicit in our dataset and have provided general contextual information. (Page 8, para 1; Pages 9 & 10)

Reviewer 2 comment:

4) The manuscript is relatively silent about gender differences, except when discussing appearance. Given the differences in survival between women and men to these ages, this is a little puzzling. Were there differences between men and women how participants expressed identities? Perhaps there weren’t, and that would be interesting in itself. But I think it bears being more explicit that in other aspects than appearance the ideas expressed by men and women were similar. This would make sense, just given the rarity of survival to these ages.

Response to reviwer 2:

4)Thank you for your feedback. There were no clear distinctions along gender lines between participants in this study, apart from refence to appearance by women. We agree that this is an important aspect that we have overlooked in the discussion. We have, therefore, added text highlighting this in the discussion (Page 40, para 1).

Reviewer 2 comment:

5)The manuscript emphasizes the relationship work these subjects do with people younger than themselves. Did the subjects discuss their relationships with similarly aged peers. Understandably these may become less important as the surviving peer group shrinks dramatically, particularly for men.

Response to reviewer 2:

5)Thank you, this is an important point which we need to be clearer about. While some participants made reference to peers, this was limited as many peers had died or were unable to visit/communicate with each other. Therefore, we were unable to identify core relationship work amongst same-aged peers. We have added a sentence to the Findings section reflecting this and a section was added to the discussion to highlight this (Page 16, para 1; Page 39, para 1)

Reviewer 2 comment:

Thanks for the opportunity to review this manuscript. I found it stimulating, and think that the paper will be of interest to a broad audience. The methodological discussion of interviewing with this group was also useful in its frank acknowledgement of the challenges

Response to reviewer 2:

Thank you for your feedback, it is much appreciated.

We also addressed the following editorial comments:

•That the manuscript follows style requirements.

We have followed the style requirements as requested.

You asked for two clarifications relating to data ethics and accessibility:

• Did participants provide their written or verbal informed consent to participate in this study? 

Yes, written consent was obtained and this has been amended in the manuscript (page 11, paragraph 1) and in the ethics statement.

• We note that you have indicated that data from this study are available upon request. PLOS only allows data to be available upon request if there are legal or ethical restrictions on sharing data publicly. In your revised cover letter, please address the following prompts:

Response:

This paper reports on a qualitative data set for a very specific group of participants (very old people aged between 97-99 years in a specific geographical setting) who could be easily identifiable, and participants did not expressly consent to their data being shared publicly. Therefore, there are ethical restrictions on the data set. While the data has been pseudonymised, the detailed nature (and select cohort) is such that it cannot be fully de-identified and consequently, is only available on request as per ethical approval from the Newcastle and North Tyneside 1 NRES committee (ref: 06/Q0905/2) (approved 2006), and subsequent substantial amendment approved by the North East - Newcastle & North Tyneside 1 Research Ethics Committee (Substantial Amendment no.23, 16th July 2019). Anonymised excerpts of the transcripts from the qualitative interviews are reported within the paper. Pseudonymised transcripts are available via the University of Newcastle Repository. Requests can be made to the Newcastle 85+ Data Repository via https://research.ncl.ac.uk/85plus/datarequests/. Formal requests for access will be considered via a data sharing agreement that indicates the criteria for data access and conditions for research use and will incorporate privacy and confidentiality standards to ensure data security. 

Branney et al (preprint DOI 10.31234/osf.io/ahdcu) suggest that open data is a continuum, rather than a binary open/closed. They, along with Prosser et al (2022) and Braun and Clarke (2021) set out why the nature of completely freely available qualitative data can be problematic, unethical, inappropriate and sometimes not desirable, in qualitative research. We have, therefore, endeavoured to make the data as open as possible within ethical restrictions in order to balance the potential risks to participants with transparency and openness.

We hope that our responses are acceptable and believe that the changes made in response to the feedback have improved our paper.

References

Peter E. Branney, Joanna Brooks, Laura Kilby, Kristina Newman, Emma Norris, Madeleine Pownall, Catherine V. Talbot, Gareth J. Treharne, Candice M. Whitaker. Three Steps to Open Science for Qualitative Research in Psychology. DOI 10.31234/osf.io/ahdcu

Annayah M.B. Prosser, Richard J. T. Hamshaw, Johanna Meyer, Ralph Bagnall, Leda Blackwood, Monique Huysamen, Abbie Jordan, Konstantina Vasileiou, Zoe Walter. When Open Data Closes the Door in Journal Submission Guidelines 1 When open data closes the door: A critical examination of the past, present and the potential future for open data guidelines in journals British Journal of Social Psychology. https://doi.org/10.1111/bjso.12576

Virginia Braun & Victoria Clarke (2021) One size fits all? What counts as quality practice in (reflexive) thematic analysis?, Qualitative Research in Psychology, 18:3, 328-352, DOI: 10.1080/14780887.2020.1769238

---

## [Decision Letter · Decision Letter 1]

27 Oct 2022

PONE-D-22-09522R1Identity construction in the very old:A qualitative narrative studyPLOS ONE

Dear Dr. Anderson,

Thank you for submitting your manuscript to PLOS ONE. After careful consideration, we feel that it has merit but does not fully meet PLOS ONE’s publication criteria as it currently stands. Therefore, we invite you to submit a revised version of the manuscript that addresses the points raised during the review process.

Many thanks for the revision that both the reviewers have appreciated. One of them still has a minor comment that I would kindly ask you to consider before I can make a final decision on publication. 

We look forward to receiving your revised manuscript.

Kind regards,

Sara Rubinelli

Academic Editor

PLOS ONE

Journal Requirements:

Reviewers' comments:

Reviewer's Responses to Questions

**Comments to the Author**

1. If the authors have adequately addressed your comments raised in a previous round of review and you feel that this manuscript is now acceptable for publication, you may indicate that here to bypass the “Comments to the Author” section, enter your conflict of interest statement in the “Confidential to Editor” section, and submit your "Accept" recommendation.

Reviewer #1: (No Response)

Reviewer #2: All comments have been addressed

2. Is the manuscript technically sound, and do the data support the conclusions?

Reviewer #1: Yes

Reviewer #2: Yes

3. Has the statistical analysis been performed appropriately and rigorously? 

Reviewer #1: N/A

Reviewer #2: N/A

4. Have the authors made all data underlying the findings in their manuscript fully available?

Reviewer #1: Yes

Reviewer #2: No

5. Is the manuscript presented in an intelligible fashion and written in standard English?

Reviewer #1: Yes

Reviewer #2: Yes

6. Review Comments to the Author

Reviewer #1: Thank you for wholeheartedly taking on board my suggestion (about the relevance of models of health)and also substantially improving the paper through other useful and thoughtful additions.

Reviewer #2: Thanks for the opportunity to read your revised manuscript, and for responding to the comments.

In particular, your revision to indicate how positioning theory has similarities to Goffman's work is just what I was looking for. I didn't intend you to redo the analysis (!), but to indicate similarities in the theoretical approach that would help readers not familiar with positioning theory place the approach.

Possibly you might signal this in the introduction in a sentence. That's the only reason I indicate minor revision. The discussion towards the end is good. Thank you.

This was a really fascinating article about an under-studied group, and I hope it will be of interest to a wide audience.

7. PLOS authors have the option to publish the peer review history of their article (what does this mean?). If published, this will include your full peer review and any attached files.

Reviewer #1: **Yes: **William Sherlaw

Reviewer #2: No

---

## [Author Response · Author response to Decision Letter 1]

8 Nov 2022

We have addressed Reviewer 2’s minor comment by adding a sentence on page 4 paragraph 2 as requested by the reviewer. We hope that our response is acceptable and would like to thank the reviewers for their positive and constructive feedback which believe have improved our paper.

---

## [Decision Letter · Decision Letter 2]

1 Dec 2022

Identity construction in the very old:A qualitative narrative study

PONE-D-22-09522R2

Dear Dr. Anderson,

We’re pleased to inform you that your manuscript has been judged scientifically suitable for publication and will be formally accepted for publication once it meets all outstanding technical requirements.

Kind regards,

Sara Rubinelli

Academic Editor

PLOS ONE

Additional Editor Comments (optional):

Reviewers' comments:

Reviewer's Responses to Questions

**Comments to the Author**

1. If the authors have adequately addressed your comments raised in a previous round of review and you feel that this manuscript is now acceptable for publication, you may indicate that here to bypass the “Comments to the Author” section, enter your conflict of interest statement in the “Confidential to Editor” section, and submit your "Accept" recommendation.

Reviewer #1: (No Response)

Reviewer #2: All comments have been addressed

2. Is the manuscript technically sound, and do the data support the conclusions?

Reviewer #1: (No Response)

Reviewer #2: Yes

3. Has the statistical analysis been performed appropriately and rigorously? 

Reviewer #1: (No Response)

Reviewer #2: N/A

4. Have the authors made all data underlying the findings in their manuscript fully available?

Reviewer #1: (No Response)

Reviewer #2: No

5. Is the manuscript presented in an intelligible fashion and written in standard English?

Reviewer #1: (No Response)

Reviewer #2: Yes

6. Review Comments to the Author

Reviewer #1: (No Response)

Reviewer #2: Thank you for your quick response to the previous queries, and congratulations on a very interesting study.

7. PLOS authors have the option to publish the peer review history of their article (what does this mean?). If published, this will include your full peer review and any attached files.

Reviewer #1: No

Reviewer #2: No

---

## [Editor Report · Acceptance letter]

6 Dec 2022

PONE-D-22-09522R2 

Identity construction in the very old:A qualitative narrative study 

Dear Dr. Anderson:

I'm pleased to inform you that your manuscript has been deemed suitable for publication in PLOS ONE. Congratulations! Your manuscript is now with our production department. 

Kind regards, 

on behalf of

Dr. Sara Rubinelli 

Academic Editor

PLOS ONE